# A Randomized Controlled Trial Investigating the Effects of Equine Simulator Riding on Low Back Pain, Morphological Changes, and Trunk Musculature in Elderly Women

**DOI:** 10.3390/medicina56110610

**Published:** 2020-11-13

**Authors:** Sihwa Park, Sunhee Park, Sukyung Min, Chang-Ju Kim, Yong-Seok Jee

**Affiliations:** 1Research Institute of Sports and Industry Science, Hanseo University, Seosan 31962, Korea; slim@korea.ac.kr (S.P.); psh1213@hanmail.net (S.P.); 102589@hanmail.net (S.M.); 2Department of Physiology, College of Medicine, Kyung Hee University, Seoul 02447, Korea; changju@khu.ac.kr

**Keywords:** equine riding simulator, back pain, elderly women, spinal alignment, isokinetic moment

## Abstract

*Background and objectives:* Studies on the effects of an equine riding simulator (ERS) program on back pain, spinal alignment, and isokinetic moments in subjects with chronic low back pain (CLBP) remain limited. The purpose of this study was to analyze changes in elderly women with CLBP who participate in an ERS program. *Materials and Methods:* The 80 participants were all women aged 61–84 years who were randomly assigned to either the control group (CON) or ERS group (ERSG). ERS exercise was performed for a duration of 12 weeks (three times each week). The degree of pain was measured using the Oswestry Disability Index and the visual analog scale. Body composition and spinal alignment were measured using bioelectrical impedance and raster stereography. The isokinetic moments of trunk extensor and flexor were measured before and after the training period. *Results:* The ERSG showed a significant decrease in back pain compared to the CON. There was a significant decrease in levels of fat in the ERSG, although no differences were shown in terms of muscle mass. However, there was an increased basal metabolic rate (BMR) in the ERSG. Spinal alignment in the ERSG significantly improved. The peak torques of the trunk extensor in the ERSG were also significantly increased. *Conclusion:* It can be inferred that the ERS exercise can decrease fat and improve the trunk extensor strength through increased BMR, leading to better spinal alignment and reducing back pain in elderly women with CLBP.

## 1. Introduction

For over 70% of the general population, low back pain (LBP) is experienced at least once in a person’s lifetime [1]. It is not an exaggeration to say that LBP becomes more chronic with age. The reason is due to the decrease in muscle mass that supports the spinal column, which acts as a pillar for the torso. When a sedentary lifestyle is added to this, LBP further worsens [2]. Physical inactivity is considered a leading risk factor that weakens the muscles connected to the spine [3]. More and more patients suffer from LBP owing to unhealthy habits, including poor posture and inactive lifestyles, especially among elderly people. According to many studies, approximately 80% of elderly people in long-term care facilities suffer significant musculoskeletal pain, with a third of them experiencing LBP. Moreover, they are not given proper treatment because their conditions are often underreported [4]. In particular, women who do not regularly exercise are more likely to suffer from LBP and disorders related to the lumbar due to a weakened musculoskeletal system [5,6]. Moreover, poor posture and pain from musculoskeletal problems lead to further deformations in the alignment of the spine. [7]. Lordosis, kyphosis, scoliosis, and pelvic torsion are deformities of normal spinal alignment. The pelvis and lumbar vertebrae alignments have effects on other parts, which can lead to more health problems. The longer LBP is left untreated, the more seriously it may affect the human body as well as the elderly. Such a condition is called chronic low back pain (CLBP) [8,9].

Among various therapeutic exercises for CLBP, equine exercise has been accepted for managing spinal cord injury or CLBP [10]. Equine exercises involve horse movements that promote the disassociation between the trunk and pelvis, leading to corrective results or postural control [11,12]. Moreover, since equine exercises can improve lumbar and pelvic joint stability, it can be considered for inclusion in exercises for core stability, as well as clinical rehabilitation programs for CLBP patients. Such exercises can enhance motor control systems and neuromuscular function for spinal injury prevention [13]. Studies have shown that equine movements have benefited patients with improved trunk postural coordination, increased motor function, and higher energy expenditure [14]. An equine riding simulator (ERS) that functions like a horse has recently been developed [15]. The use of ERS has been shown to result in increased muscular contraction [16] and the improvement of balance, mobility, and gait pattern, which could be induced by hormonal and cerebral activity changes in older adults [17]. Another study has shown improvement in an upright posture by triggering particular muscles responsible for such maintenance [18]. Additionally, continuous adjustments, mainly due to stronger abdominal, lumbar, and pelvic muscles, led to an improvement in postural balance and control [19,20].

However, few studies have investigated the application of ERS for treating elderly CLBP patients. ERS may benefit patients with CLBP, but information is lacking regarding its possible pain reduction improvement in the trunk musculature and in morphological properties, including body composition and spinal alignment in the elderly. From what we know, there have been no studies that have evaluated the ability of a systematic ERS program to affect pain, trunk muscle contractions, and spinal alignment responsible for lumbosacral stability. Thus, we performed a single-blind randomized controlled trial on elderly women with CLBP for studying the effects of an ERS program.

## 2. Materials and Methods

### 2.1. Patients

Eighty patients with CLBP voluntarily participated in this randomized controlled study. This study followed the principles of the Declaration of Helsinki and received approval from the institutional ethics committee (2 October 2018 to 1 October 2019; 2-7001793-AB-N-012018116HR). Before enrolling in the study, all subjects signed an informed consent form. The subjects were elderly women, with a mean age of 71.77 ± 6.55 years, residing in Seoul Seniors Tower in Korea, who were interested in using an ERS. This study specifically sought to recruit individuals with CLBP who did not exercise regularly for at least 6 months. An individual was diagnosed as having CLBP if LBP persisted for 6 months or more with no known pathological basis, such as neurological causes or LBP induced by trauma. Women who underwent any kind of operation involving the spine had severe musculoskeletal disorders, or cardiovascular complications were excluded. Having dementia, which may make it difficult to answer the questionnaire; having received treatment or taking any medication that can affect physical conditions; having fibromyalgia syndromes, or any contraindication to rehabilitation (infection, fever, hypothyroidism, cancer, or other systemic diseases) were also reasons for exclusion [18]. Any anti-inflammation medication, pain relievers, anticoagulants, muscle relaxants, or antidepressants taken one week before the initiation of the study were also grounds for exclusion. Before the study started, the details of all the procedures were thoroughly explained to the subjects who were then asked to answer questionnaires, including questions on demographics, visual analog scale (VAS), and the Oswestry Disability Index (ODI).

Initially, 84 participants were screened to determine their study eligibility. Throughout the course of the study, one subject had to be excluded due to personal reasons. Another subject in the equine riding simulator group (ERSG) quit during the assessment phase, and two subjects in the control (CON) group failed to remain until the follow-up phase. In the end, 80 subjects took part in this research. An expert explained that the subjects had to sit on the horse simulator while watching a screen for 30 min a day for 12 weeks (3 times each week), but they did not know if the simulator could move until allotting the groups. Then, they were assigned using random number tables and given identification numbers upon recruitment. The participants were randomly allocated to either CON (*n* = 40) or ERSG (*n* = 40). To inhibit communication between ERSG and CON, the subjects were classified according to their community areas where the ERSG was sent to a rehabilitation laboratory in the morning and the CON in the afternoon. At the beginning of the program, only the ERSG realized that the horse simulator moved. The complete study characteristics are presented in Table 1.

### 2.2. Experimental Design

All participants answered a self-reported questionnaire that included VAS and ODI, which measure the degrees of pain experienced in the lower back when being in a comfortable posture to an active posture [21]. The ERS program was developed based on previous research [15,22]. The equestrian simulator (FORTIS 101, Daewon, Corp., Seoul, Korea) was 1.7 m in height, 315 kg in weight, 580 mm in width, 1900 mm in length, and was designed to mechanically reproduce the movement of an actual horse. The ERS exercise was modified and supplemented to suit the participants of this research. The ERSG was trained by an equestrian simulator 30 min per session, 3 days per week, for 12 weeks. While the ERSG participants were exercising on the horse, the CON participants sat on the horse and watched the video from the monitor.

The ERS program included 8 min of warm-ups, 15 min of exercise, and 7 min of cool-downs. To prevent injuries, all participants underwent stretching while standing before the intervention and learned how to ride the equine simulator. After boarding the simulator, participants walked for 3 min as a pre-exercise before the actual exercise. During the ERS program, researchers watched participants from the side and helped them to establish the correct posture. After the actual exercise, participants walked again for 2 min and finished with stretching on the ground.

Generally, the exercise intensity of ERS depends on the horse’s movement (i.e., footwork in a real horse), which is classified as walking, trotting, cantering, and galloping. A horse’s walking pattern is in the form of a four-beat walking speed of 6 km/h in the beginning. Before the walking speed increases, the order of a horse’s leg movements follows the pattern of: left hind leg–left front leg–right hind leg–right front leg. As the speed reaches the critical point, it changes to two-beats at a speed of 15 km/h, in which the diagonal legs of the horse move simultaneously. Trotting is classified into sitting and rising trots. A sitting trot involves trotting with the hips attached to the saddle, whereas a rising trot involves lifting the buttocks in the saddle to match the upper and lower rhythm of the horse, making the horse run with more liveliness by reducing the bodyweight of the rider during the trotting. As the speed increases, the cantering with a velocity of 24 km/h changes from an unbalanced triplet movement to galloping; that is, the maximum speed of galloping is about 60 km/h. This study applied an intensity between walking and cantering in consideration of the balance ability and stability of elderly women. First, participants began walking on the horse simulator for 15 min during Weeks 1–4. The goal of ordinary walking was to adapt to the machine and horse-movement patterns. During Weeks 5–8, participants walked for 10 min to trotting for 5 min on the horse simulator. The goal of this period was to improve neuromuscular control and maintain balance. During Weeks 9–12, participants performed trotting for 10 min to cantering for 5 min on the horse simulator. The goal of this period was to strengthen the major muscles around the spine, pelvis, and legs. The rehabilitation programs and the duration of the rides in this study’s program were based on previously published research [15,23,24] and used for managing CLBP conditions and preventing reinjury [25].

### 2.3. Back Pain Measurement

To measure the level of pain, the study used VAS to identify variables related to movements of the body, pain during the night, pain while lying down or standing, and tension in the waist. The subjects rated their back pain with a bipolar rating scale. The scale was shown in a rectangular box (10 cm × 5 cm). The range started with no pain at “0” to intense pain at “10”. The subjects marked their level of pain along the scale and a numerical score was obtained by placing a transparent sheet over it. The subjects then took the ODI questionnaire in which a score was based on 100 percentage points [21]. The ODI is often used to measure conditions for spinal disorders [26,27] and includes ten parts that measure pain levels and the degree to how it interferes with various kinds of physical activities. The subjects took the ODI that was translated into Korean when they had a consultation with their physician. The sex life question was omitted in the Korean version of the ODI. The questionnaire is a reliable and valid way to assess the ability to function for individuals with spinal disorder [28]. The subjects were given an evaluation by psychologists at the start and end of the study [29]. Cronbach’s α, representing internal consistency, was calculated to measure the reliability of the ODI. The Cronbach’s α of the VAS and ODI in this study was 0.877 and 0.885, respectively.

### 2.4. Isokinetic Moments’ Measurements

An isokinetic dynamometer (HUMAC^®^/NORM^TM^ Testing and Rehabilitation System, CSMi, Stoughton, MA, USA) was used in accordance with the protocol for assessing trunk extension and flexion (TEF) [30]. In the TEF test, the subjects placed their feet on the footplate of the TEF module with the heels against the heel cups. The height of the footage was coordinated until the length of the rubber alignment was 3.5 cm below the top of the iliac crest for the alignment of their vertical axis with a dynamometer to adjust. The pelvic belt was secured around the upper anterior superior iliac spines. The height of the popliteal pad was positioned to the rear of the patella and the lower body was held in place with the popliteal, tibial, and thigh pads with a slight bend in the knees. The backs of the subjects were placed on the sacral pad in which fore and aft movements were made by the alignment wheel so that the pointer was centered at the axis of rotation. To prevent the lower limb from protruding forward, the legs were secured in place for conducting the TEF test. The leg was secured in place by two pads that aligned the lower and upper parts of the leg to the top and bottom parts of the patella. After aligning these pads, the lever was locked. To secure the upper body in a fixed position, a pad was attached to the chest as the subjects grasped the handles for added stability. The bottom part of the chest pad was made to match the inferior angle of the scapula and positioned parallel to the scapula pad. The subjects completed 4 maximal warm-up trials and 5 maximal test trials at 30°/s and 60°/s, respectively, which gained a peak torque. Next, the subjects completed 4 maximal warm-up trials and 15 maximal test trials at 120°/s, which gained a peak torque. 60 s of rest were given between tests under the supervision of one expert researcher.

### 2.5. Morphological Measurements

Firstly, for body composition measurements, Inbody 230 (Biospace Co., Ltd., Seoul, Korea) was used with the bioelectrical impedance analysis method. The subjects removed metal objects, socks, or other articles that could interfere with electric currents prior to getting on the device. Subjects held the handles and stood stationary for 3 min. Before each assessment, the subjects refrained from eating, drinking, and exercising for 4 h, 12 h, and 7 days, respectively. Subjects also voided 30 min before each assessment [29]. Secondly, the vertebral formation is three-dimensional (3D) and consists of coronal, sagittal, and transverse planes. Therefore, a 3D image analyzer was used for examining the spinal alignment. A Formetric III 4D (Diers International GmbH, Schlangenbad, Germany) was used to photogrammetrically record the spinal shape with a video raster stereography process. This provided a precise 3D image of the back surface [31,32]. It projects uniform lines horizontally on the backs of patients. When the light raster is observed from a point that differs from the projection, distortions of the lines reveal data about the shape. Since the lines of light are simultaneously projected and registered all at once, the amount of time to measure the surface is approximately 0.25 s. The Formetric III 4D performs automatic localization of anatomical landmarks on the back surface. An example of this would be the spinous process of the C7 vertebra (vertebra prominent (VP)) or the left and right dimple points in the pelvic area (posterior superior iliac spine) shown in Figure 1. Using these points, a sagittal profile and an outline of the back are produced [33].

All patients were asked to step on the plate, take off their shirts, lower their underwear so that the tail bone was visible, and stand in a natural position with the arms at the sides for taking pictures of the back. Through this process, trunk imbalance (slope of the trunk), trunk inclination, pelvic tilt, pelvic torsion, surface rotation, trunk rotation, lateral deviation or spine curvature, kyphotic angle of the thoracic vertebrae, and lordotic angle of the lumbar vertebrae were assessed.

### 2.6. Statistical Analyses

Microsoft Excel (Microsoft, Redmond, WA, USA) was used to analyze the data, expressed as mean ± standard deviation (SD). The sample size was determined using G*Power v. 3.1.9.4, considering a priori effect size of f^2^ (*V*) = 0.295 (medium size effect), α error probability = 0.05, and power (1-β error probability) = 0.95. A sample size of 80 was recommended, and the current sample of this study included 80 participants. The effect size was calculated according to Cohen’s *d,* which is equal to the mean difference of the groups divided by the pooled SD. SPSS program (version 22.0; IBM Corp., Armonk, NY, USA) was used to perform all statistical analyses and the Shapiro–Wilk test was used to check data distribution. Differences between the groups were observed using the Mann–Whitney U test prior to comparing the groups. Analysis of variance (ANOVA) test was used for evaluating significant variances between the groups at baseline, and 2 × 2 (group, time, and group by time interaction) was used to assess the effects of intervention. Analysis of covariance (ANCOVA) test was used to determine the difference between groups if there was an interaction between group and time (pre and post). An intention-to-treat analysis was conducted to compare ERSG and CON. ERSG vs. CON served as the between-group factor and the baseline vs. Week 12 was the within-group factor. Δ% was obtained through additional analysis of variables between times. For all analyses, the significance level was set at *p* ≤ 0.05.

## 3. Results

### 3.1. Effect of ERS Exercise on Back Pain

As shown in Table 2, the VAS and ODI scores in the CON did not improve but showed significant changes in the ERSG. The Δ% of the VAS score in the CON increased by 5.06%, whereas that in the ERSG decreased by 71.45% (data not shown). The Δ% of ODI score in the CON increased by 3.93%, whereas that in the ERSG decreased by 59.79% (data not shown). The ANCOVA showed that the VAS and ODI scores in ERSG were significantly reduced at post-time compared with CON (F = 184.428, *p* < 0.001 and F = 370.083, *p* < 0.000, respectively). These changes indicate that the effects of ERS exercise led to more significant relief of back pain for activities done on a daily basis.

### 3.2. Effect of ERS Exercise on Trunk Extensor and Flexor

As shown in Table 3, although all isokinetic moments at all angular velocities in the CON tended to decrease, these variables in the ERSG increased. Specifically, the Δ% of the trunk extensor at 30°/s in the CON decreased by 13.66% but increased by 7.14% in the ERSG (data not shown). Although the Δ% of the trunk flexors at 120°/s in the CON decreased by 2.61%, it increased by 14.78% in the ERSG (data not shown). The Δ% of the trunk extensor at 60°/s and 120°/s in the CON decreased by 12.91% and 13.51%, respectively, and that in the ERSG increased by 3.09% and 30.15%, respectively (data not shown). The ANCOVA also showed that variables of the trunk extensor at 30°/s, 60°/s, and 120°/s in ERSG were significantly higher than those in CON (F = 13.247, *p* < 0.000, F = 5.761, *p* = 0.019, and F = 20.757, *p* < 0.000, respectively). Moreover, ERS exercise significantly increased the trunk flexor at 120°/s (F = 15.311, *p* < 0.000). Therefore, ERS exercise was shown to involve a higher amount of exercise, which can strengthen the contractile properties of the trunk extensor. Moreover, there were significant interactions at all angular velocities.

### 3.3. Effect of ERS Exercise on Body Composition

As shown in Table 4, baseline factors in the subjects of both CON and ERSG showed homogeneity. The groups showed different results, except for muscle mass. Body weight, fat mass, and fat percentage in the CON increased, whereas those in the ERSG decreased. The ANCOVA revealed that variables of body weight, fat mass, and fat percentage in ERSG were significantly lower than those in CON (F = 15.348, *p* < 0.000, F = 19.376, *p* < 0.000, and F = 13.750, *p* < 0.000, respectively), and BMR in ERSG was significantly higher than that in CON (F = 16.663, *p* < 0.000). This shows that ERS exercise can lead to a reduction of fat levels with no changes in muscle mass in elderly women. Moreover, ERS exercise effectively facilitated the basal metabolic rate.

### 3.4. Effect of ERS Exercise on Spinal Alignment

There were some interactions between group and time (pre and post) as shown in Table 5. Therefore, we performed ANCOVA. Trunk inclination and imbalance in CON were similar between pre- and post-time. On the other hand, trunk inclination and imbalance in ERSG were significantly decreased compared to CON (F = 14.210, *p* < 0.001 and F = 10.925, *p* = 0.001, respectively). Such tendency was found in pelvic tilt, kyphotic angle, and lordotic angle, and their differences were statistically significant (F = 13.364, *p* < 0.001, F = 35.881, *p* < 0.001, and F = 11.994, *p* = 0.001, respectively). The ERS exercise significantly affected the surface rotation to the right side and the lateral deviation to the left side of the trunk compared to CON (F = 14.214, *p* < 0.001 and F = 5.070, *p* = 0.027, respectively).

## 4. Discussion

The results of this research showed the benefits of ERS exercise on elderly women with CLBP. Upon completing 12 weeks of ERS, back pain significantly decreased and the trunk extensor strength increased in the elderly women. Furthermore, some of the body composition and spinal alignment of the ERSG showed improvement. In other words, it is estimated that the 12-week ERS program can reduce back pain and improve body composition and spinal alignment by causing positive changes in the trunk extensor strength in elderly women with CLBP.

Several studies have reported the application of exercise for treating CLBP. Our findings are in line with many previous studies [10,11,12,13,14,15]. Relatively, considering that the subjects are the elderly, the experiment period seems to be lengthy. This study observed positive changes in VAS and ODI scores in elderly CLBP patients. There were distinct differences between groups through ANCOVA analysis. In contrast to the CON that showed no significant difference in back pain, the ERSG showed significant reductions in back pain scores (VAS, 71.45%; ODI, 59.79%) after 12 weeks. A possible explanation for the decreased level of back pain may be due to performing stability exercises on the equine simulator that applied deep muscle stimulation and stretching for improved spinal alignment. Manniche et al. [33] used an LBP rating scale for 105 participants with CLBP. They used intensive dynamic back exercises for CLBP with a total of 30 sessions for 12 weeks. Although the low- and high-intensity exercise groups performed latissimus pull-down and back extension exercises, the control group only exercised lightly on the floor. The LBP rating scale was used to assess disability, pain, and physical impairment. By the end of their experiment, the results from the LBP rating scale showed that the exercise group improved considerably more than the other groups. Furthermore, Risch et al. [34] reported the effects of lumbar strengthening exercise once or twice a week for 10 weeks in participants with CLBP. The results of their study showed that lumbar extension exercise is beneficial for strengthening the lumbar extensors and results in decreased pain and improved perceptions of physical and psychosocial functioning in CLBP patients. However, these improvements were not related to changes in psychological distress. In other words, it can be inferred that physiological changes reduced their back pain. Hodges [13] also stated that exercise programs for treating CLBP should increase neuromuscular function and build the supporting muscles for the pelvis and spine. Akuthota et al. [35] emphasized the importance of core stability for a balanced distribution of force on the pelvis and spine. Other authors also reported that exercises for core stability can benefit the trunk muscles and contribute to spinal stabilization, coordination, and control [36]. Similarly, ERS used in our study may create a core stability effect, as reported in previous studies [37]. However, there has been no research regarding the effects of ERS on mechanical muscle contractions and spinal alignment of the abdomen and back, which are associated with lumbopelvic stability and postural structure. Moreover, there has been a lack of studies investigating whether ERS can reduce back pain in the elderly with CLBP.

The ERS exercise program in our study resulted in positive changes in the muscle contractions of the trunk. Specifically, the trunk extensor in this study showed improvement, which was measured by an isokinetic device, as reported in previous studies [38,39]. In detail, the results from the ANCOVA test in this study showed that variables of the trunk extensor strengths at 30°/s, 60°/s, and 120°/s in ERSG were significantly higher than those in CON. Moreover, ERS exercise significantly increased the trunk flexor strength at 120°/s after 12 weeks of intervention. It is reasonable for individuals with CLBP to engage in proper exercise when considering the results of improved endurance and the ability to mitigate back pain [40,41]. This study showed that 12 weeks of ERS exercise resulted in a significant muscle contraction difference at 120°/s when compared to the start of the intervention. In regards to back pain, a significant difference was found after 12 weeks in VAS and ODI between the groups when compared to the baseline. These results point to the benefits that ERS exercise can have on increasing muscular endurance and reducing back pain. Kennedy and Noh [42] conducted a study that showed how strength deficits can be corrected through a comprehensive rehabilitation program by having subjects subsequently progress to functional exercises. The authors advocated exercise as a way to strengthen the back. One of the purposes of spinal rehabilitation programs is to improve and strengthen the lower back [43].

Regular exercise is critical for reducing body weight and is the most important habit to adopt for individuals with LBP [44]. Roffey et al. [45] and Thompson et al. [46] reported that exercising regularly can decrease subcutaneous adipose tissue and reduce visceral adiposity in patients with LBP, which is consistent with our findings that the effects of ERS exercise can decrease fat in elderly women with CLBP. In other words, ERS exercise can improve body composition in individuals with CLBP and improve the muscles surrounding the lumbar joints. A population-based study found that moderate and vigorous physical activities were closely tied to a greater chance of persistent LBP for women ≥65 years of age, whereas walking for 30 min for at least five days a week and doing strength exercises for at least two days a week decreased the risk of persistent LBP when accounting for differences in age and body mass index [47,48]. In addition, ERS exercise may provide deeper muscle activation in the moving horse simulator and generate a greater metabolic energy consumption effect compared to the CON, though the subjects were moving while the ERS remained in place. Hence, ERS exercise can contribute to reducing fat by improving metabolic effectiveness and strengthening the paraspinal muscles by mechanical muscle contraction. Similarly, the results from the ANCOVA test in this study revealed that variables of body weight, fat mass, and % fat in ERSG were significantly lower than those in CON, and BMR in ERSG was significantly higher than that in CON. This means that ERS exercise can lead to a reduction of fat levels with no changes in muscle mass in elderly women. In other words, it can be presumed that ERS exercise improves trunk extensor strength, resulting in a change in the BMR, contributing to fat mass reduction but not enhancing muscle mass due to age.

It has also been reported that ERS exercise can induce a decrease in fat mass through an increase in the trunk, and helps maintain a good spinal alignment [49]. Similarly, positive changes through ERS exercise from the ANCOVA test in this study were observed in the trunk inclination and imbalance, pelvic tilt, kyphotic and lordotic angles, surface rotation to the right side, as well as lateral deviation to the left side after 12 weeks. It was revealed that the indexes of spinal alignment in elderly women who underwent the ERS program for 12 weeks were close to within the normal standard deviation of most indicators related to spinal alignment [50]. Surprisingly, although the participants in our study did not have severe spinal misalignment, the CON without ERS tended to deviate from the normal spinal alignment range. In contrast, the vertebral alignment status of the elderly who underwent ERS exercise was hardly far from the normal range. This could be caused by changes in the muscle function required for erecting and moving the spine. There are studies that have shown improved muscle strength and contraction in elderly individuals from using a riding simulator [16]. Some studies revealed improved changes in posture, which affect certain muscle groups for retaining posture against gravity [18]. Continuous changes have been reported to strengthen the muscles in the pelvis, abdomen, and lumbar for enhanced trunk balance and control of posture [19]. Moreover, considerable attention has been observed in the past decade on sagittal alignment and balance in individuals who have spinal complications [51]. The quantifying pelvic incidence and practical usefulness of analyzing spinal alignment/balance for clinical purposes have been identified by several studies [52,53,54]. However, no significant changes were observed in all variables related to spinal alignment through ERS exercise in this study. In other words, the surface rotation to the left side and lateral deviation to the right side of the trunk did not show a significant difference in the interaction after 12 weeks of ERS exercise. This may include not only the reason that the subjects of this study are older adults but also the failure to observe a spinal alignment through equipment such as computerized tomography or magnetic resonance imaging. In addition, it would be desirable to observe changes in the disability of CLBP patients through the fingertip-to-floor-test along [55] with advanced testing equipment.

Upon completion of the study, the lumbar exercise group showed a significant improvement over the control group in pain intensity, as well as lumbar extensor strength. Previous studies have also indicated that sufficient exercise in LBP patients can effectively decrease pain and reduce disability [56,57,58]. Moreover, Rainville et al. [59] explained that LBP patients do not have to be concerned about their safety when exercising since it does not contribute to a greater risk of injuring their backs. Other researchers suggested the possibility of performing progressive core stability exercises to help reduce discomfort in women with CLBP [35]. Implementing such intervention programs that involved specific types of exercise appear to be helpful as a study by Koumantakis et al. [5] showed that eight weeks of core stability training reduced pain levels. Suggesting that individuals with CLBP to perform lumbar stabilization exercises as part of a more comprehensive intervention program may be conducive to mitigating pain as well as enhancing blood flow, preventing spasms, and reducing inflammation [60,61]. The ERS exercise program that was used in this study can offer similar effects of performing the core stability exercises that were implemented as intervention programs for CLBP. When taking into consideration that the horse simulator movements contributed to the balancing of both sides of the paraspinal muscles, ERS exercise may be a useful tool for those with structural deformations. However, our elderly participants did not have severe issues with their skeletal structure, such as a lordotic or kyphotic or scoliotic spine. Therefore, there should be careful consideration when applying ERS exercise to the elderly with spinal deformities. In other words, it would be desirable to apply ERS to elderly patients after a detailed observation of the characteristics of each CLBP patient.

## 5. Conclusions

This study concluded that ERS exercise can improve back pain through enhancing the trunk extensor at all angular velocities and the trunk flexor at 120°/s and through improving morphological properties such as reducing fat and correcting some spinal alignment for elderly women with CLBP. These effects were seen in participants who chronically complained of LBP owing to the neural tube restriction because of muscle and ligament deformations surrounding the lumbosacral joint. Performing ERS exercise for a duration of 12 weeks was shown to strengthen the paraspinal muscles through stimulating the lumbosacral joint.

### Limitations

However, our study has some limitations. First, the rater or evaluator was not blinded to group allocation. Second, the participants consisted entirely of women with a small sample size. Third, this study used a 4D analyzer, which is less accurate than a CT or MRI in observing the spinal alignment of CLBP patients. Considering these limitations, further studies that compare the effectiveness of exercising with a real horse with that of a horse simulator are encouraged.

## Figures and Tables

**Figure 1 medicina-56-00610-f001:**
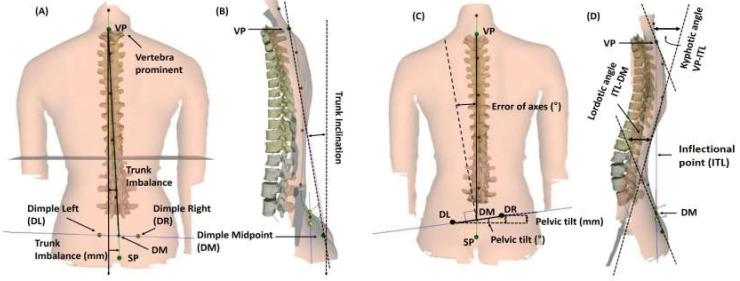
Raster stereographic analysis of spinal alignment. (**A**) Trunk imbalance, which was calculated by vertebra prominent (VP), dimple midpoint (DM), dimple left (DL), and dimple right (DR). (**B**) Trunk inclination. (**C**) Pelvic tilt and torsion (error of axis), which could calculate the surface rotation and lateral deviation. (**D**) Kyphotic angle between VP and the inflectional point (ITL) and the lordotic angle between ITL and DM.

**Table 1 medicina-56-00610-t001:** Physical characteristics of the participants at baseline.

Variables (Unit)	Groups		
	CON (*n* = 40)	ERSG (*n* = 40)	*p*-value
Age (y)	72.05 ± 6.82	71.50 ± 6.34	0.732
Height (cm)	159.96 ± 7.92	161.30 ± 8.74	0.410
Weight (kg)	64.03 ± 9.38	60.23 ± 6.88	0.363
Muscle mass (kg)	42.75 ± 6.91	42.98 ± 5.72	0.063
Fat mass (kg)	18.44 ± 4.20	17.41 ± 4.94	0.348
Percent fat (%)	27.23 ± 5.01	27.35 ± 6.16	0.840
LBP history (month)	22.10 ± 7.47	23.61 ± 8.47	0.550

All data represent the mean ± standard deviation. LBP, low back pain; CON, control group; ERSG, equine riding simulator group. *p*-value was analyzed using the Mann–Whitney U test.

**Table 2 medicina-56-00610-t002:** Comparison of back pain severity.

Items	Time (T)	Group (G)		ANOVA (*p*-Value)		
		CON (*n* = 40)	ERSG (*n* = 40)	G	T	G × T
VAS	Pre	7.27 ± 1.52	7.35 ± 1.63	0.001	0.001	0.001
	Post	7.64 ± 1.31	2.10 ± 2.54			
ODI	Pre	43.98 ± 6.58	44.31 ± 6.68	0.001	0.001	0.001
	Post	45.71 ± 8.04	17.82 ± 4.66			

All data represent the mean ± standard deviation. CON, control group; ERSG, equine riding simulator group; VAS, visual analog scale; ODI, Oswestry Disability Index; ANOVA, analysis of variance. *p*-value was analyzed using a repeated-measures ANOVA test.

**Table 3 medicina-56-00610-t003:** Comparative results of isokinetic peak torques from trunk extensor and flexor.

Items (Units)	Time (T)	Group (G)		ANOVA (*p*-Value)		
		CON (*n* = 40)	ERSG (*n* = 40)	G	T	G × T
Trunk extensor at 30°/s (Nm)	Pre	120.03 ± 48.36	128.20 ± 42.13	0.030	0.373	0.002
	Post	103.63 ± 49.62	137.35 ± 44.23			
Trunk flexor at 30°/s (Nm)	Pre	140.03 ± 43.45	139.73 ± 28.08	0.486	0.441	0.061
	Post	136.55 ± 45.63	147.98 ± 32.62			
Trunk extensor at 60°/s (Nm)	Pre	112.30 ± 38.27	117.25 ± 47.37	0.133	0.203	0.036
	Post	97.80 ± 43.58	120.88 ± 51.46			
Trunk flexor at 60°/s (Nm)	Pre	128.90 ± 46.87	145.38 ± 35.82	0.052	0.782	0.842
	Post	127.83 ± 43.02	145.20 ± 31.28			
Trunk extensor at 120°/s (Nm)	Pre	79.20 ± 39.17	83.00 ± 33.03	0.015	0.073	0.001
	Post	68.50 ± 41.46	108.03 ± 54.56			
Trunk flexor at 120°/s (Nm)	Pre	102.35 ± 46.71	124.50 ± 39.67	0.001	0.011	0.001
	Post	99.68 ± 48.26	142.90 ± 43.66			

All data represent the mean ± standard deviation. CON, control group; ERSG, equine riding simulator group; ANOVA, analysis of variance. *p*-value was analyzed using a repeated-measures ANOVA test.

**Table 4 medicina-56-00610-t004:** Comparative results of participants’ body composition.

Items (Units)	Time (T)	Group (G)		ANOVA (*p*-Value)		
		CON (*n* = 40)	ERSG (*n* = 40)	G	T	G × T
Weight (kg)	Pre	64.03 ± 9.38	61.42 ± 7.90	0.061	0.314	0.002
	Post	64.64 ± 9.04	60.23 ± 6.88			
Muscle mass (kg)	Pre	42.75 ± 6.91	42.98 ± 5.72	0.667	0.002	0.123
	Post	41.65 ± 6.64	42.60 ± 6.06			
Fat mass (kg)	Pre	18.44 ± 4.20	17.41 ± 4.94	0.009	0.305	0.001
	Post	20.38 ± 4.80	16.31 ± 4.52			
Percent fat (%)	Pre	27.23 ± 5.01	27.35 ± 6.16	0.080	0.001	0.001
	Post	31.62 ± 6.16	27.33 ± 6.43			
BMR (kcal/day)	Pre	1261.30 ± 149.25	1238.09 ± 127.86	0.962	0.048	0.001
	Post	1248.47 ± 145.06	1274.58 ± 129.66			

All data represent the mean ± standard deviation. CON, control group; ERSG, equine riding simulator group; BMR, basal metabolic rate; ANOVA, analysis of variance. *p*-value was analyzed using a repeated-measures ANOVA test.

**Table 5 medicina-56-00610-t005:** Comparative results of participants’ spinal alignment.

Items (Units)	Time (T)	Group (G)		ANOVA (*p*-Value)		
		CON (*n* = 40)	ERSG (*n* = 40)	G	T	G × T
Trunk inclination (°)	Pre	2.27 ± 1.25	2.41 ± 1.37	0.126	0.016	0.001
	Post	2.40 ± 1.22	1.54 ± 1.14			
Trunk imbalance (°)	Pre	5.84 ± 2.21	5.87 ± 2.32	0.167	0.030	0.005
	Post	5.99 ± 1.79	4.74 ± 2.36			
Pelvic tilt (°)	Pre	4.63 ± 1.41	4.68 ± 1.62	0.075	0.117	0.001
	Post	4.95 ± 1.61	3.79 ± 1.67			
Pelvic torsion (°)	Pre	2.21 ± 0.91	2.26 ± 0.97	0.323	0.532	0.076
	Post	2.35 ± 0.81	1.97 ± 0.96			
Kyphotic angle (°)	Pre	45.58 ± 8.15	44.26 ± 7.79	0.001	0.006	0.001
	Post	46.88 ± 7.87	36.57 ± 7.26			
Lordotic angle (°)	Pre	39.61 ± 8.45	38.63 ± 8.73	0.017	0.045	0.004
	Post	40.63 ± 9.80	33.03 ± 10.32			
Surface rotation to the right side (°)	Pre	4.17 ± 2.13	4.13 ± 1.95	0.103	0.504	0.005
	Post	4.61 ± 1.56	3.43 ± 1.78			
Surface rotation to the left side (°)	Pre	−4.07 ± 2.56	−4.17 ± 2.52	0.481	0.253	0.169
	Post	−4.14 ± 2.35	−3.41 ± 2.28			
Lateral deviation to the right side (mm)	Pre	5.80 ± 2.00	6.02 ± 2.20	0.766	0.523	0.202
	Post	5.97 ± 2.05	5.51 ± 2.28			
Lateral deviation to the left side (mm)	Pre	−5.42 ± 2.03	−5.63 ± 1.97	0.210	0.375	0.028
	Post	−5.86 ± 2.58	−4.60 ± 2.81			

All data represent the mean ± standard deviation. CON, control group; ERSG, equine riding simulator group; ANOVA, analysis of variance. *p*-value was analyzed using a repeated-measures ANOVA test.

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
