# Peer review of "A Randomized Controlled Trial Investigating the Effects of Equine Simulator Riding on Low Back Pain, Morphological Changes, and Trunk Musculature in Elderly Women"

_medicina, 2020, doi:10.3390/medicina56110610_

Round 1

Reviewer 1 Report

This is an interesting study. I have some major concerns but I think the authors will be able to address them:

1. Title: Morpholomechanical or morpho-mechanical?

2. Title: Please, identify the article as “randomized controlled trial”

3. Abstract: Please, identify the article as “randomized controlled trial”

4. Introduction: Missing information about the benefits of horse riding simulators on chronic pain (see a recent review “Effects of Equine-Assisted Therapies or Horse-Riding Simulators on Chronic Pain: A Systematic Review and Meta-Analysis”) and also about the known benefits in the elderly.

5. Experimental design: please, try to explain better the procedure. I find it confusing. What is “walking on the horse simulator”?

6. Results: line 202, 212 and 224, please, start reporting the GxT differences. Between-group differences are the most interesting differences in Randomized controlled trials and those differences must be highlighted.

7. Discussion: Why starting with the differences in body composition? Please, start with the most relevant outcome (pain levels).

8. Discussion: “significant changes in the degree of pain could be attributed to ERS exercise because of decreased body fat, correction of spinal alignment, and development of the trunk extension muscles.” I think this statement is not based on your results. Maybe you can say that there could be a link between them (you could explore the relations between the improvements using your data), but for me, as it is, it is not accurate.

9. Line 272: “Moreover, there have been no studies that have investigated whether ERS can reduce back pain in individuals with CLBP”. That is not true, see the article “Time-effects of horse simulator exercise on psychophysiological responses in men with chronic low back pain” by Hye-Won Oh et al.

10. Fat mass may be an interesting variable but, can you differentiate between areas? I mean, the potential link between fat mass and LBP in elderly women will not probably be the same if they lose fat mass from the lower limb or from the core...

11. Limitations must be reported in the discussion section. If the study protocol was not previously, it must be declared as limitation.

Author Response

Answers to 1st reviewer’s comments

Thank you for your kind advice and comments for publication in Medicina. We revised our manuscript as per your comments. We represented the specific modifications in response to the comments by blue-letters in our manuscript. We sincerely appreciate your comments because your comments make our manuscript better. Details of responses about reviewer’ comments are as follows.

#1. Comments or Suggestions

Title: Morpholomechanical or morpho-mechanical?

#1. Response: Thank you for what you pointed out. We will accept whatever you are asked to choose the title. However, after reconsidering our manuscript, we would like to choose the title as follows: “Equine Simulator Riding Improves Back Pain and Mechanical or Morphological Property in Elderly with CLBP: A Randomized Controlled Trial”.

Moreover, we changed from Abstract to Conclusion according to the Title.

For example, “Abstract: Background and objectives: Studies on the effects of an equine riding simulator (ERS) program on back pain and mechanical property (isokinetic moments) or morphological property (body composition and spinal alignment) in subjects with chronic low back pain (CLBP) remain limited. The purpose of this study was to analyse changes in elderly women with CLBP who participate in an ERS program. Materials and Methods: The 80 participants were all women aged 61–84 years who were randomly assigned to either the control group (CON) or ERS group (ERSG) for a randomised controlled trial. ERS exercise was performed for a duration of 12 weeks (3 times each week). The degree of pain was measured using the Oswestry Disability Index and the visual analog scale. The isokinetic moments for the mechanical properties of trunk extensor and flexor were measured before and after the training period. Body composition and spinal alignment for morphological property were measured using bioelectrical impedance and raster stereography. Results: The ERSG showed a significant decrease in back pain compared to the CON. In mechanical property, the peak torques of the trunk extensor in the ERSG were significantly increased. In morphological property, there was a significant decrease in levels of fat in the ERSG, although no differences were shown in terms of muscle mass. Spinal alignment in the ERSG significantly improved. Conclusion: It can be inferred that the ERS exercise can reduce back pain through increasing the trunk extensor and leading to better fatness and spinal alignment in elderly women with CLBP.”

#2. Comments or Suggestions

Title: Please, identify the article as “randomized controlled trial”

#2. Response: Thank you for what you pointed out. According to your suggestion, we inserted the phrase into the end of the Title such as “Equine Simulator Riding Improves Back Pain and Mechanical or Morphological Property in Elderly with CLBP: A Randomized Controlled Trial”.

#3. Comments or Suggestions

Abstract: Please, identify the article as “randomized controlled trial”

#3. Response: Thank you for what you pointed out. Considering your suggestion, we changed the sentences and inserted these as follows:

Line 17-18 of the modified manuscript version: “…..ERS group (ERSG) for a randomised controlled trial.”

#4. Comments or Suggestions

Introduction: Missing information about the benefits of horse riding simulators on chronic pain (see a recent review “Effects of Equine-Assisted Therapies or Horse-Riding Simulators on Chronic Pain: A Systematic Review and Meta-Analysis”) and also about the known benefits in the elderly.

#4. Response: Thank you for what you pointed out. Considering your suggestion, we changed the sentences and inserted the two references as follows:

Line 49-51 of the modified manuscript version: “Equine exercises involve horse movements that promotes the disassociation between the trunk and pelvis, which can lead to corrective results or to postural control [11, 12].”

Line 57-59 of the modified manuscript version: “The use of ERS has been shown to result in increased muscular contraction [16] and to improvement of balance, mobility, and gait pattern which could induced by hormonal and cerebral activity changes in elderly people [17]”

[12] Collado-Mateo, D.; Lavín-Pérez, A. M.; García, J. P. F.; García-Gordillo, M. Á.; Villafaina, S. Effects of Equine-Assisted Therapies or Horse-Riding Simulators on Chronic Pain: A Systematic Review and Meta-Analysis. Medicina (Kaunas). 2020, 56, E444. doi: 10.3390/medicina56090444.

[17] Hilliere, C.; Collado-Mateo, D.; Villafaina, S.; Duque-Fonseca, P.; Parraca, J.A. Benefits of Hippotherapy and Horse Riding Simulation Exercise on Healthy Older Adults: A Systematic Review. PM&R 2018, 10, 1062–1072. doi.org/10.1016/j.pmrj.2018.03.019.

Of course, we have also modified the order of the references below the conclusion. Thank you for your suggestion.

#5. Comments or Suggestions

Experimental design: please, try to explain better the procedure. I find it confusing. What is “walking on the horse simulator”?

#5. Response: Thank you for what you pointed out. Considering your suggestion, we can explain the locomotion of the equine simulator. The locomotion of the equine simulator was made according to the movement and speed of the walk, trot, canter and gallop that the actual horse moves. According to your suggestion, we inserted the sentences as follows:

Line 114-126 of the modified manuscript version: “Generally, the exercise intensity at ERS depends on the horse's movement (i.e. footwork in a real horse) which are classified with walking, trot, canter, and gallop. Horse's walking pattern is in the form of a four-beat walking at a speed of 6 km/h in the beginning. Before the walking speed increases, the order of walking performs left hind leg - left front leg - right hind leg - right front leg. As the speed reaches the critical point, it changes to a two-beat breaking at a speed of 15 km/h, and diagonal legs of the horse simultaneously move. Trot is classified as sitting trot and rising trot. Sitting trot is the trotting with the hips attached to the saddle, whereas rising trot is the lifting of the buttocks in the saddle to match the upper and lower rhythm of the horse. It makes the horse run more lively by reducing the body weight of the rider during the trotting. As the speed increases, the canter with a velocity of 24 km/h, changes from unbalanced triplet movement to gallop. That is, the maximum speed of the gallop is about 60 km/h. This study applied from walking to cantering as the intensity of exercise in consideration of the balance ability and stability of elderly women.”

#6. Comments or Suggestions

Results: line 202, 212 and 224, please, start reporting the GxT differences. Between-group differences are the most interesting differences in Randomized controlled trials and those differences must be highlighted.

#6. Response: Thank you for what you pointed out. According to your suggestion, we inserted the sentences as follows:

Line 216-217 of the modified manuscript version: “There were no significant differences between the groups in the VAS (Z = -0.376; P = 0.707) and in the ODI (Z = -0.337; P = 0.736) prior to the experiment. As shown in Table 2, ….”

Line 228-230 of the modified manuscript version: “There were no significant differences between the groups in body weight (Z = -0.909; P = 0.363), muscle mass (Z = -0.529; P = 0.597), fat mass (Z = -0.938; P = 0.348), percent fat (Z = -0.202; P = 0.840), and BMR (Z = -0.500; P = 0.617) prior to the experiment. As shown in Table 3, ….”

Line 242-247 of the modified manuscript version: “There were no significant differences between the groups in trunk inclination (Z = -0.525; P = 0.600), trunk imbalance (Z = -0.308; P = 0.758), pelvic tilt (Z = -0.120; P = 0.904), pelvic torsion (Z = -0.039; P = 0.969), kyphotic angle (Z = -0.857; P = 0.392), lordotic angle (Z = -0.409; P = 0.682), surface rotation to right side (Z = -0.053; P = 0.958), surface rotation to left side (Z = -0.096; P = 0.923), lateral deviation to right side (Z = -0.448; P = 0.654), and lateral deviation to left side (Z = -0.486; P = 0.627) prior to the experiment. As shown in Table 4, ….”

Line 255-258 of the modified manuscript version: “There were no significant differences between the groups in trunk extensor at 30°/s (Z = -0.712; P = 0.476), trunk flexor at 30°/s (Z = -0.029; P = 0.977), trunk extensor at 60°/s (Z = -0.096; P = 0.923), trunk flexor at 60°/s (Z = -1.473; P = 0.141), trunk extensor at 120°/s (Z = -0.245; P = 0.806), and trunk flexor at 120°/s (Z = -1.757; P = 0.079) prior to the experiment. As shown in Table 5, ….”

#7. Comments or Suggestions

Discussion: Why starting with the differences in body composition? Please, start with the most relevant outcome (pain levels).

#7. Response: Thank you for what you pointed out. According to your suggestion, we changed the order of outcome from back pain to morphological variables. In addition, we changed the order of references in the sentences as follows:

Line 271-304 of the modified manuscript version: “…Upon completing 12 weeks of ERS, the back pain of the elderly women significantly decreased and the trunk extensor of them significantly increased. Furthermore, the morphological property including body composition and spinal alignment of the ERSG showed that a positive effect that can be improved. In other words, it is estimated that the 12-week ERS program can reduce pain and improve morphological properties by causing positive changes in mechanical functions in the elderly women with CLBP.

Several studies have reported the application of exercise for treating CLBP. Our findings are in line with many previous studies [10-15]. Due to the long-term intervention period of this study, it was expected that there would be positive changes in the VAS and ODI scores for back pain in CLBP patients. In contrast to the CON that showed no significant difference in back pain, ERSG showed significant reductions in back pain scores (VAS, 71.45%; ODI, 59.79%) after 12 weeks. A possible explanation for the decreased level of back pain may be due to performing stability exercises on the equine simulator that applied deep muscle stimulation and stretching for improved spinal alignment. Manniche et al. [33] used an LBP rating scale for 105 participants with CLBP. They used intensive dynamic back exercises for CLBP with a total of 30 sessions for 12 weeks. Although the low and high intensity exercise groups performed latissimus pull down and back extension exercises, the control group only exercised lightly on the floor. The LBP rating scale was use to assess disability, pain, and physical impairment. By the end of their experiment, the results from the LBP rating scale showed that the exercise group improved considerably more than the other groups. Furthermore, Risch et al. [34] reported the effects of lumbar strengthening exercise for once or twice a week for 10 weeks in participants with CLBP. The results of their study showed that lumbar extension exercise is beneficial for strengthening the lumbar extensors and results in decreased pain and improved perceptions of physical and psychosocial functioning in CLBP patients. However, these improvements were not related to changes in psychological distress. In other words, it can be inferred that the back pain was reduced by physiological changes. Hodges [13] stated that exercise programs for treating CLBP should increase neuromuscular function and build the supporting muscles for the pelvis and spine. Akuthota et al. [35] emphasized the importance of core stability for a balanced distribution of force on the pelvis and spine. Noormohammadpour et al. [36] also reported that exercises for core stability can benefit the trunk muscles and contribute to spinal stabilisation, coordination, and control. Similarly, ERS used in our study may create a core stability effect as reported in previous studies [37]. However, there has been no research regarding the effects of ERS on mechanical muscle contractions and spinal alignment of the abdomen and back, which are associated with lumbopelvic stability. Moreover, there have been lacking studies that have investigated whether ERS can reduce back pain in the elderly with CLBP….”

#8. Comments or Suggestions

Discussion: “significant changes in the degree of pain could be attributed to ERS exercise because of decreased body fat, correction of spinal alignment, and development of the trunk extension muscles.” I think this statement is not based on your results. Maybe you can say that there could be a link between them (you could explore the relations between the improvements using your data), but for me, as it is, it is not accurate.

#8. Response: Thank you for what you pointed out. According to your suggestion, we changed the sentences based on our results of this study as follows:

Line 271-274 of the modified manuscript version: “Upon completing 12 weeks of ERS, the back pain of the elderly women significantly decreased and the trunk extensor of them significantly increased. Furthermore, the morphological property including body composition and spinal alignment of the ERSG showed that a positive effect that can be improved.”

#9. Comments or Suggestions

Line 272: “Moreover, there have been no studies that have investigated whether ERS can reduce back pain in individuals with CLBP”. That is not true, see the article “Time-effects of horse simulator exercise on psychophysiological responses in men with chronic low back pain” by Hye-Won Oh et al.

#9. Response: Thank you for what you pointed out. According to your suggestion, we changed the sentences based on our results of this study as follows:

Line 303-304 of the modified manuscript version: “Moreover, there have been lacking studies that have investigated whether ERS can reduce back pain in the elderly with CLBP.”

#10. Comments or Suggestions

Fat mass may be an interesting variable but, can you differentiate between areas? I mean, the potential link between fat mass and LBP in elderly women will not probably be the same if they lose fat mass from the lower limb or from the core...

#10. Response: Thank you for what you pointed out. As in the Line 318 and 319 of the modified manuscript version, we tried to separate the potential link between fat mass and LBP in elderly women. In other words, we suggested that the pain and mechanical property wrote on Line 277-318 as follows.

“Several studies have reported the application of exercise for treating CLBP. Our findings are in line with many previous studies [10-15]. Due to the long-term intervention period of this study, it was expected that there would be positive changes in the VAS and ODI scores for back pain in CLBP patients. In contrast to the CON that showed no significant difference in back pain, ERSG showed significant reductions in back pain scores (VAS, 71.45%; ODI, 59.79%) after 12 weeks. A possible explanation for the decreased level of back pain may be due to performing stability exercises on the equine simulator that applied deep muscle stimulation and stretching for improved spinal alignment………”

Moreover, we suggested that the morphological property wrote under Line 319 as follows.

“In the aspect of morphological property, a regular exercising is critical for reducing body weight and is the most important habit to adopt for individuals with LBP [44]. Roffey et al. [45] and Thompson et al. [46] reported that exercising regularly can decrease subcutaneous adipose tissue and reduce visceral adiposity in patients with LBP, which is consistent with our findings that the effects of ERS exercise can decrease fat in elderly women with CLBP……”

#11. Comments or Suggestions

Limitations must be reported in the discussion section. If the study protocol was not previously, it must be declared as limitation.

#11. Response: Thank you for what you pointed out. According to your suggestion, we changed the sentences based on our results of this study as follows:

Line 369-373 of the modified manuscript version: “However, our elderly participants did not have severe issues with their skeletal structure, such as a lordotic or kyphotic or scoliotic spine. Therefore, it is thought that there will be limitations in applying ERS to the elderly with spinal deformity. In other words, it would be desirable to apply the ERS to the elderly patients after thorough observation of the characteristics of each CLBP patient.”

Re-submission Date

25 October 2020

Reviewer 2 Report

General and specific comments to medicina-977012 entitled „Equine simulator riding improves back pain and morpholomechanical property in elderly women with chronic low back pain“ (Park J et al.).

This is a well written report of a clinical study on equine simulator riding on chronic low back pain in elderly women.

Several points should be addressed and discussed:

  1. Line 78 to 84: Exclusion criteria have been described. As horse riding and the according transfers may have an impact on bone stability, osteoporosis could have been a reason for exclusion. Have patients been screened for osteoporosis?
  2. Line 87 and line 189 to 199: How large were effect sizes assumed? Was a sample size calculation done?
  3. Line 84 to 87: Please provide more information about the informed consent. How was horse riding explained? This is most important as the control group sat on the horse while there was no movement from it. The question is whether “watching the video” has been accepted as “real” horse riding by the participants. What was the content of the video? Did the active intervention group watched the same video, too? The study was designated as “single-blind”. Have any credibility tests been done?
  4. Line 120 to 188: Among the different outcomes. What was the primary outcome? What were secondary outcome parameters? These specifications are most important to get a correctly powered sample size (see point 2).
  5. Line 144: For examining the spinal alignment the Formetric III 4D analyser was used. Please explain the (pseudo-)dynamic advantage of the 4D analysis in comparison to the previous 3D model measure (e.g. see in Frerich et al.).
  6. Line 214 to 215: “This shows …. While increasing muscle mass in elderly women. This is not shown by data. Therefore, this sentence is misleading. Please correct.
  7. Line 228, Table 4: Changes in 4D analysis seem to be very subtle. Please explain (in the discussion) how these little changes might have clinical importance. Please provide some references for this. The reference Frerich et al. (41) (line 281) did not show the correlation between the changes in spine alignment and clinical parameters (pain, mobility etc..). The Frerich study focused on the correlation between 4D analysis and results from standard radiographs in scoliosis. Please correct this. In contrast, Ekedahl et al. (Arch Phys Med Rehabil 2012;93:2210-2215) showed that the fingertip-to-floor-test (FTF) has a good validity in low back pain as a predictor of changes in disability. Has the FTF-test been done? The probably unproven/weak correlation of the 4D analysis and clinical outcomes in CLBP may also be mentioned in the limitation section.
  8. Line 325: It should be “used”.
  9. The discussion should be presented in a more structured way.

Author Response

Answers to 2nd reviewer’s comments

Thank you for your kind advice and comments for publication in Medicina. We revised our manuscript as per your comments. We represented the specific modifications in response to the comments by blue-letters in our manuscript. We sincerely appreciate your comments because your comments make our manuscript better.

#1. Comments or Suggestions

Line 78 to 84: Exclusion criteria have been described. As horse riding and the according transfers may have an impact on bone stability, osteoporosis could have been a reason for exclusion. Have patients been screened for osteoporosis?

#1. Response: Thank you for what you pointed out. The participants living in Seoul Seniors Tower in Korea receive health checkups every six months. The elderly had osteopenia with a T-score of -1 to -2.5, but no osteoporosis. Even with osteoporosis, there were only a few elderly people who took medication for osteoporosis. Therefore, we inserted only ‘severe musculoskeletal disorders’ in our manuscript. However, we think the reviewer's opinion seemed to be correct, we inserted 'osteoporosis' in the manuscript on Line 83 of the modified manuscript version.

#2. Comments or Suggestions

Line 87 and line 189 to 199: How large were effect sizes assumed? Was a sample size calculation done?

#2. Response: Thank you for what you pointed out. We calculated the effect size through G*Power v 3.1.9.4. The Cohen's d is the appropriate effect size measure if two groups have similar standard deviations and are of the same size. Glass's delta, which uses only the standard deviation of the control group, is an alternative measure if each group has a different standard deviation. Hedges' g, which provides a measure of effect size weighted according to the relative size of each sample, is an alternative where there are different sample sizes. In addition, if the different sample sizes, we should use Hedges' g. We selected Cohen's d to test the effect size of the dependent variable to be observed. The reason is that in this study two groups have similar standard deviations and are of the same size. We referred to the method of calculating the sample size of de Bekker-Grob EW et al (2015). The number of subjects was selected according to the method. [de Bekker-Grob EW et al. (2015). Sample Size Requirements for Discrete-Choice Experiments in Healthcare: a Practical Guide. Patient. 8(5):373-84. doi: 10.1007/s40271-015-0118-z.] Considering to your opinion, we inserted the sentences related to sample size and effect sizes to the 'Statistical analyses’ of this study as follows:

Line 213-217 of the modified manuscript version: “The sample size was determined using G*Power v 3.1.9.4, considering an a priori effect size f2(V) = 0.295 (medium size effect), α error probability = 0.05, and power (1-β error probability) = 0.95. A sample size of 80 was recommended, and the current sample of this study included 80 participants. Effect size was calculated according to Cohen’s d (which is equal to the mean difference of the groups divided by the pooled SD)”

#3. Comments or Suggestions

Line 84 to 87: Please provide more information about the informed consent. How was horse riding explained? This is most important as the control group sat on the horse while there was no movement from it. The question is whether “watching the video” has been accepted as “real” horse riding by the participants. What was the content of the video? Did the active intervention group watched the same video, too? The study was designated as “single-blind”. Have any credibility tests been done?

#3. Response: Thank you for what you pointed out. According to your suggestion, we can apply as follows: From the beginning of this study, we did not advertise to all participants about the ERS program. We just advertised to sit on a horse simulator and watch a video about health information. We also explained that all subjects should watch health-related series for 30 minutes three days a week. In the case of the CON, an expert helped to avoid bending back while sitting on the horse simulator. On the other hand, in the case of the ERSG, an expert assisted and observed to perform from walking to cantering of ERS for 12 weeks while sitting on a horse simulator. Recruitment of subjects, advertisements, and management methods of different groups are considered to have no problem with reliability because the following references are applied. [Ref.] Kim, J.; Jee,Y. EMS-effect of Exercises with Music on Fatness and Biomarkers of Obese Elderly Women. Medicina (Kaunas). 2020, 56(4):158. doi: 10.3390/medicina56040158.

Meanwhile, referring to your comments, we put the details in our manuscript as follows:

Line 76-78 of the modified manuscript version: “Before enrolling in the study, all subjects signed an informed consent, which advertised to sit on a horse simulator and to watch health-related series for 30 minutes three days a week for 12 weeks.” ............

Line 90-98 of the modified manuscript version: “Before the study started, all patients knew they had to sit on the horse simulator when watching a video which related to health information for 30 min a day, but they did not know if the simulator could be moved until separating the control group (CON) and the equine riding simulator group (ERSG). Then, they were assigned using random number tables and assigned identification numbers upon recruitment. In order to prevent communication between the CON who was not provided with a simulation movement and the ERSG who could be provided with a simulation movement, the patients were classified according to their community areas, and the ERSG was sent to a rehabilitation center in the morning, while the CON in the afternoon. At the beginning of the program, only the ERSG realized that a horse simulator was moving........”

#4. Comments or Suggestions

Line 120 to 188: Among the different outcomes. What was the primary outcome? What were secondary outcome parameters? These specifications are most important to get a correctly powered sample size (see point 2).

#4. Response: Thank you for what you pointed out. According to your suggestion, we changed the orders from the primary outcome to secondary outcome parameters on Line 145-210 of the modified manuscript version.

#5. Comments or Suggestions

Line 144: For examining the spinal alignment the Formetric III 4D analyser was used. Please explain the (pseudo-)dynamic advantage of the 4D analysis in comparison to the previous 3D model measure (e.g. see in Frerich et al.).

#5. Response: Thank you for what you pointed out. We summarized it as follows: “Frerich et al. (2012) investigated to the Formetric 4D surface topography system was compared to standard radiography as a safer option for evaluating patients with adolescent idiopathic scoliosis (AIS). Fourteen volunteers with typical AIS patient stature had 30 repeated Formetric 4D measurements taken, and reproducibility was assessed. Sixty-four patients with AIS were then enrolled during routine clinic visits. Evaluation included standard radiographs and surface topography measurements. A comparison analysis was performed in their test methods. Ultimately, they reported that the Formetric 4D is comparable to radiography in terms of its reproducibility. Although the Formetric 4D does not predict curve magnitude exactly, the predictions correlate strongly with the Cobb angles determined from radiographs.”

According to your suggestion, we deleted the vague sentences and only objective facts were inserted in this study as follows:

Line 191-192 of the modified manuscript version: “This provided a precise 3D image of the back surface. Up to 500 X-ray images that were digitized and analysed, were used to make comparisons for confirming the reliability of the Formetric III 4D. [31,32].”

#6. Comments or Suggestions

Line 214 to 215: “This shows …. While increasing muscle mass in elderly women. This is not shown by data. Therefore, this sentence is misleading. Please correct.

#6. Response: Thank you for what you pointed out. According to your suggestion, we changed the sentences based on our results of this study as follows:

Line 260-261 of the modified manuscript version: “This shows that ERS exercise can lead to a reduction of fat levels while no changing muscle mass in elderly women.”

#7. Comments or Suggestions

Line 228, Table 4: Changes in 4D analysis seem to be very subtle. Please explain (in the discussion) how these little changes might have clinical importance. Please provide some references for this. The reference Frerich et al. (41) (line 281) did not show the correlation between the changes in spine alignment and clinical parameters (pain, mobility etc..). The Frerich study focused on the correlation between 4D analysis and results from standard radiographs in scoliosis. Please correct this. In contrast, Ekedahl et al. (Arch Phys Med Rehabil 2012;93:2210-2215) showed that the fingertip-to-floor-test (FTF) has a good validity in low back pain as a predictor of changes in disability. Has the FTF-test been done? The probably unproven/weak correlation of the 4D analysis and clinical outcomes in CLBP may also be mentioned in the limitation section.

#7. Response: Thank you for what you pointed out. According to your advice, we changed the sentences based on our results of this study as follows:

Line 274-282 of the modified manuscript version: “As shown in Table 5, the trunk inclination and trunk imbalance of the CON tended to increase, while those of ERSG tended to decrease after the end of 12-week experiment. These results indicated a considerable difference in group × time interaction (P < 0.05). These tendencies were similar in pelvic tilt, but did not show any significant change in the pelvic torsion. Kyphotic angle and lordotic angle also showed significant changes in in group × time interaction.

Meanwhile, since the surface rotation to right side of the trunk increased in CON while it decreased in ERSG, there was a significant change in the interaction. The remaining three variables, surface rotation to left side, lateral deviation to right side, and lateral deviation to left side, showed no statistically significant differences or changes after 12 weeks.”

Also, we inserted the sentences the limitation about unproven/weak correlation of the 4D analysis and clinical outcomes in CLBP as follows.

Line 376-382 of the modified manuscript version: “However, in this study, no significant changes were observed in all variables related to spinal alignment through ERS exercise. In other words, the surface rotation to left side (P = 0.169) and lateral deviation to right side (P = 0.202) of the trunk did not show a significant difference in the interaction after 12 weeks of ERS exercise. The reason for this may include not only the reason that the subject of this study is an elderly person, but also the failure to observe spinal alignment through equipment such as computerized tomography or magnetic resonance imaging.”

#8. Comments or Suggestions

Line 325: It should be “used”.

#8. Response: Thank you for what you pointed out. We corrected the word as follows:

Line 304 of the modified manuscript version: “......The LBP rating scale was used to assess disability, pain, and physical impairment....”

#9. Comments or Suggestions

The discussion should be presented in a more structured way.

#9. Response: Thank you for what you pointed out. According to your suggestion, we changed the Discussion based on our results of this study as follows:

Line 288-403 of the modified manuscript version:

“The results of this research showed the benefits of ERS exercise on elderly women with CLBP. Upon completing 12 weeks of ERS, the back pain significantly decreased and the trunk extensor of the mechanical property increased in the elderly women. Furthermore, some of the morphological property including body composition and spinal alignment of the ERSG showed that a positive effect that can be improved. In other words, it is estimated that the 12-week ERS program can reduce back pain and improve morphological properties by causing positive changes in mechanical functions in the elderly women with CLBP.

Several studies have reported the application of exercise for treating CLBP. Our findings are in line with many previous studies [10-15]. Due to the long-term intervention period of this study, it was expected that there would be positive changes in the VAS and ODI scores for back pain in CLBP patients. In contrast to the CON that showed no significant difference in back pain, the ERSG showed significant reductions in back pain scores (VAS, 71.45%; ODI, 59.79%) after 12 weeks. A possible explanation for the decreased level of back pain may be due to performing stability exercises on the equine simulator that applied deep muscle stimulation and stretching for improved spinal alignment. Manniche et al. [33] used an LBP rating scale for 105 participants with CLBP. They used intensive dynamic back exercises for CLBP with a total of 30 sessions for 12 weeks. Although the low and high intensity exercise groups performed latissimus pull down and back extension exercises, the control group only exercised lightly on the floor. The LBP rating scale was used to assess disability, pain, and physical impairment. By the end of their experiment, the results from the LBP rating scale showed that the exercise group improved considerably more than the other groups. Furthermore, Risch et al. [34] reported the effects of lumbar strengthening exercise for once or twice a week for 10 weeks in participants with CLBP. The results of their study showed that lumbar extension exercise is beneficial for strengthening the lumbar extensors and results in decreased pain and improved perceptions of physical and psychosocial functioning in CLBP patients. However, these improvements were not related to changes in psychological distress. In other words, it can be inferred that the back pain was reduced by physiological changes. Hodges [13] also stated that exercise programs for treating CLBP should increase neuromuscular function and build the supporting muscles for the pelvis and spine. Akuthota et al. [35] emphasized the importance of core stability for a balanced distribution of force on the pelvis and spine. Another authors also reported that exercises for core stability can benefit the trunk muscles and contribute to spinal stabilisation, coordination, and control [36]. Similarly, ERS used in our study may create a core stability effect as reported in previous studies [37]. However, there has been no research regarding the effects of ERS on mechanical muscle contractions and spinal alignment of the abdomen and back, which are associated with lumbopelvic stability and postural structure. Moreover, there have been lacking studies that have investigated whether ERS can reduce back pain in the elderly with CLBP.

The ERS exercise program in our study resulted in positive changes in the muscle contractions of the trunk. Specifically, the trunk extensor in this study showed improvement, which was measured by an isokinetic device, as reported in previous studies [38,39]. Although the trunk flexor in the ERSG seemed to increase, those of the CON showed the opposite. However, these results indicated no significant difference in group × time interaction, except for the trunk flexor at 120°/s. Unlike these results, this study found that the trunk extensor of mechanical property in the ERSG at all angular velocities (30°/s, 60°/s, and 120°/s) significantly increased after 12 weeks of intervention. It is reasonable for individuals with CLBP to engage in proper exercise when considering the results of improved endurance and the ability to mitigate back pain [40,41]. This study showed that 12 weeks of ERS exercise resulted in a significant muscle contraction differences at 120°/s when compared to the start of the intervention. In regards to back pain, a significant difference was found after 12 weeks in VAS and ODI between the groups when compared to the baseline. These results point to the benefits that ERS exercise can having on increasing muscular endurance and reducing back pain. Kennedy and Noh [42] conducted a study that showed how strength deficits can be corrected through a comprehensive rehabilitation program by having subjects subsequently progress to functional exercises. The authors advocated exercise as a way to strength the back. One of the purposes of spinal rehabilitation programs is to improve and strengthen the lower back [43].

In the aspect of morphological property, a regular exercising is critical for reducing body weight and is the most important habit to adopt for individuals with LBP [44]. Roffey et al. [45] and Thompson et al. [46] reported that exercising regularly can decrease subcutaneous adipose tissue and reduce visceral adiposity in patients with LBP, which is consistent with our findings that the effects of ERS exercise can decrease fat in elderly women with CLBP. In other words, the ERS exercise can improve body composition in individuals with CLBP and improve the muscles surrounding the lumbar joints. A population-based study found that moderate and vigorous physical activities were closely tied to a greater chance of persistent LBP for women ≥ 65 years of age, whereas walking for 30 mins for at least 5 days a week and doing strength exercises for at least 2 days a week decreased the risk of persistent LBP when accounting for differences in age and body mass index [47,48]. In addition, ERS exercise may provide deeper muscle activation in the moving horse simulator and generate a greater metabolic energy consumption effect compared to the CON, though the subjects were moving while the ERS remained in place. Hence, ERS exercise can contribute to reducing fat by improving metabolic effectiveness and strengthening the paraspinal muscles by mechanical muscle contraction. However, this study did not find that ERSG increased muscle mass through ERS exercise, which is thought to be related to the mean age of 71.77 ± 6.55 years. In other words, it can be presumed that ERS exercise improves mechanical muscle contraction ability, resulting in a change in metabolic rate, contributing to fat mass reduction, but not enhancing muscle mass due to age increase.

It has also been reported that ERS exercise can induce a decrease in fat mass through an increase in the mechanical contraction capacity of the trunk, and also helps maintain a good spinal alignment [49]. Similarly, positive changes through ERS exercise in this study were observed in the trunk inclination and imbalance, pelvic tilt, kyphotic and lordotic angles, surface rotation to the right side, as well as lateral deviation to the left side after 12 weeks. It was revealed that the indexes of spinal alignment in elderly women who underwent the ERS program for 12 weeks were close to within the normal standard deviation of most indicators related to spinal alignment [50]. Surprisingly, although the participants in our study did not have severe spinal misalignment, the CON without ERS tended to deviate from the normal spinal alignment range. In contrast, the vertebral alignment status of the elderly who underwent ERS exercise was hardly far from the normal range. This could be caused by changes in the mechanical muscle function required for erecting and moving the spine. There are studies that have shown improved muscle strength and contraction in elderly individuals from using a riding simulator [16]. Some studies revealed improved changes in posture which affect certain muscle groups for retaining posture against gravity [18]. Continuous changes have been reported to strengthen the muscles in the pelvis, abdomen, and lumbar for enhanced trunk balance and control of posture [19]. Moreover, considerable attention has been observed in the past decade on sagittal alignment and balance in individuals who have spinal complications [51]. The quantifying pelvic incidence and practical usefulness of analysing spinal alignment/balance for clinical purposes have been identified by several studies [52-54]. However, in this study, no significant changes were observed in all variables related to spinal alignment through ERS exercise. In other words, the surface rotation to left side (P = 0.169) and lateral deviation to right side (P = 0.202) of the trunk did not show a significant difference in the interaction after 12 weeks of ERS exercise. The reason for this may include not only the reason that the subject of this study is an elderly person, but also the failure to observe a spinal alignment through equipment such as computerized tomography or magnetic resonance imaging. In addition, it would be desirable to observe changes in the disability of CLBP patients through the fingertip-to-floor-test along [55] with advanced testing equipment.

Upon completion of the study, the lumbar exercise group showed a significant improvement over the control group in pain intensity, as well as lumbar extensor strength. Previous studies have also indicated that sufficient exercise in LBP patients can effectively decrease pain and reduce disability [56–58]. Moreover, Rainville et al. [59] explained that LBP patients do not have to be concerned about their safety when exercising since it does not contribute to a greater risk of injuring their backs. Other researchers suggested the possibility of performing progressive core stability exercises to help reduce discomfort in women with CLBP [35]. Implementing such intervention programs that involved specific types of exercise appear to be helpful as a study by Koumantakis et al. [5] showed that 8 weeks of core stability training reduced pain levels. Suggesting that individuals with CLBP to perform lumbar stabilization exercises as part of a more comprehensive intervention program may be conducive in mitigating pain as well as enhancing blood flow, preventing spasms, and reducing inflammation [60,61]. The ERS exercise program that was used in this study can offer similar effects of performing the core stability exercises that were implemented as intervention programs for CLBP. When taking into consideration that the horse simulator movements contributed to the balancing of both sides of the paraspinal muscles, ERS exercise may be a useful tool for those with structural deformations. However, our elderly participants did not have severe issues with their skeletal structure, such as a lordotic or kyphotic or scoliotic spine. Therefore, it is thought that there will be considered in applying ERS exercise to the elderly with spinal deformity. In other words, it would be desirable to apply the ERS to the elderly patients after detail observation of the characteristics of each CLBP patient.”

Submission Date

25 October 2020

Round 2

Reviewer 1 Report

Most of my comments have been addressed. However, I think the authors have not interpreted well the results. In the last review, when I suggested to start with the between group differences, I was talking about the interaction Group*Time, and not the between group differences at baseline. In randomized controlled trials, the interaction GxT compares the changes in the two groups, and that is the main result, not the baseline differences or the within-group differences. We need to know if the improvements in the experimental group are higher than those observed in the control group. Please, focus in the GxT results, both in the results and in the discussion section. 

Author Response

Answers to 1st reviewer’s comments

Thank you for your kind advice and comments for publication in Medicina. We revised our manuscript as per your comments. We represented the specific modifications in response to the comments by blue-letters in our manuscript. We sincerely appreciate your comments because your comments make our manuscript better. Details of responses about reviewer’ comments are as follows.

#1. Comments or Suggestions

Title: Most of my comments have been addressed. However, I think the authors have not interpreted well the results. In the last review, when I suggested to start with the between group differences, I was talking about the interaction Group*Time, and not the between group differences at baseline. In randomized controlled trials, the interaction GxT compares the changes in the two groups, and that is the main result, not the baseline differences or the within-group differences. We need to know if the improvements in the experimental group are higher than those observed in the control group. Please, focus in the GxT results, both in the results and in the discussion section. 

#1. Response: Thank you for what you pointed out. We carried out the two-way repeated measures ANOVA to compare the means of variables between group and time (pre and post). As you pointed out, there were some interaction Group * Time in some variables such as VAS, ODI and Trunk extensor at 30°/s. According to your comment, we need to identify that the improvements in the experimental group are higher than those observed in the control group. However, the interaction between group and time can affect the misinterpretation of our results. Therefore, we performed analysis of covariance (ANCOVA) of the variables at pre-time with the variables at pre-time as covariate. We inserted this content into “2.6. Statistical analyses” and “Results”. In addition, we changed the sentences in the Discussion referring to new results.

For examples,

On Line 213 to 228:

2.6. Statistical analyses

 Microsoft Excel (Microsoft, Redmond, WA, USA) was used to analyse the data, expressed as mean ± standard deviation (SD). The sample size was determined using G*Power v3.1.9.4, considering a priori effect size of f2(V) = 0.295 (medium size effect), α error probability = 0.05, and power (1-β error probability) = 0.95. A sample size of 80 was recommended, and the current sample of this study included 80 participants. The effect size was calculated according to Cohen’s d, which is equal to the mean difference of the groups divided by the pooled SD. SPSS program (version 22.0; IBM Corp., Armonk, NY, USA) was used to perform all statistical analyses and the Shapiro–Wilk test was sued to check data distribution. Differences between the groups were observed using the Mann–Whitney U test prior to comparing the groups. Analysis of variance (ANOVA) test was used for evaluating significant variances between the groups at baseline. 2 × 2 (group, time, and group by time interaction) was used to assess the effects of intervention. Analysis of covariance (ANCOVA) test was used to determine the difference between groups if there was the interaction between group and time (pre and post). An intention-to-treat analysis was conducted to compare ERSG and CON. ERSG vs. CON served as the between-group factor and the baseline vs. Week 12 was the within-group factor. Δ% was obtained through additional analysis of variables between times. For all analyses, the significance level was set at P ≤ 0.05.

On Line 231 to 280:

3.1. Effect of ERS Exercise on Back Pain

As shown in Table 2, the VAS and ODI scores in the CON did not improve, but showed significant changes in the ERSG; the Δ% of the VAS score in the CON increased by 5.06%, whereas that in the ERSG decreased by 71.45% (data not shown). The Δ% of ODI score in the CON increased by 3.93%, whereas that in the ERSG decreased by 59.79% (data not shown). The analysis of covariance (ANCOVA) showed that the VAS and ODI scores in ERSG were significantly reduced at post-time compared with CON (F = 28.746, P < 0.001 and F = 8.088, P < 0.000, respectively). These changes indicate that the effects of ERS exercise led to a more significant relief of back pain for activities done on a daily basis.

3.2. Effect of ERS Exercise on Trunk Extensor and Flexor of Mechanical Property

As shown in Table 3, although all isokinetic moments at all angular velocities in the CON tended to decrease, these variables in the ERSG increased. Specifically, the Δ% of the trunk extensor at 30°/s in the CON decreased by 13.66%, but increased by 7.14% in the ERSG (data not shown). Although the Δ% of the trunk flexors at 120°/s in the CON decreased by 2.61%, it increased by 14.78% in the ERSG (data not shown). The Δ% of the trunk extensor at 60°/s and 120°/s in the CON decreased by 12.91% and 13.51%, respectively, and that in the ERSG increased by 3.09% and 30.15%, respectively (data not shown). The ANCOVA also showed that variables of the trunk extensor at 30°/s, 60°/s, and 120°/s in ERSG were significantly higher than those in CON (F = 13.247, P < 0.000, F = 5.761, P = 0.019, and F = 20.757, P < 0.000, respectively). Moreover, ERS exercise significantly increased the trunk flexor at 120°/s (F = 15.311, P < 0.000). Therefore, ERS exercise was shown to involve a higher amount of exercise, which can strengthen the contractive properties of the trunk extensor. Moreover, there were significant interactions at all angular velocities.

3.3. Effect of ERS Exercise on Body Composition of Morphological Property

As shown in Table 4, baseline factors in the subjects of both CON and ERSG showed homogeneity. The groups showed different results, except for muscle mass. Body weight, fat mass, and fat percentage in the CON increased, whereas those in the ERSG decreased. The ANCOVA revealed that variables of body weight, fat mass, and fat percentage in ERSG were significantly lower than those in CON (F = 15.348, P < 0.000, F = 19.376, P < 0.000, and F = 13.750, P < 0.000, respectively), and BMR in ERSG was significantly higher than that in CON (F = 16.663, P < 0.000). This shows that ERS exercise can lead to a reduction of fat levels with no changes in muscle mass in elderly women. Moreover, ERS exercise effectively facilitated the basal metabolic rate.

3.4. Effect of ERS Exercise on Spinal Alignment of Morphological Property

There were some interactions between group and time (pre and post) as shown in Table 5. Therefore, we performed ANCOVA. Trunk inclination and imbalance in CON were similar between pre- and post-time. On the other hand, trunk inclination and imbalance in ERSG were significantly decreased compared to CON (F = 14.210, P < 0.001 and F = 10.925, P = 0.001, respectively). Such tendency was found in pelvic tilt, kyphotic angle, and lordotic angle, and their differences were statistically significant (F = 13.364, P < 0.001, F = 35.881, P < 0.001, and F = 11.994, P = 0.001, respectively). The ERS exercise significantly affected the surface rotation to right side and the lateral deviation to left side of the trunk compared to CON (F = 14.214, P < 0.001 and F = 5.070, P = 0.027, respectively).

  1. Discussion

................. On Line 292 to 296:

" Our findings are in line with many previous studies [10-15]. Relatively, considering that the subjects are the elderly, the experiment period seems to be not a short period. This study observed positive changes in VAS and ODI scores in elderly CLBP patients, and found that there were distinct differences between groups through ANCOVA analysis.”

................. On Line 323 to 326:

" In detail, the results from ANCOVA test in this study showed that variables of the trunk extensor at 30°/s, 60°/s, and 120°/s in ERSG were significantly higher than those in CON. Moreover, ERS exercise significantly increased the trunk flexor at 120°/s after 12 weeks of intervention.”

................. On Line 351 to 356:

" Similarly, the results from ANCOVA test in this study revealed that variables of body weight, fat mass, and % fat in ERSG were significantly lower than those in CON, and BMR in ERSG was significantly higher than that in CON. This means that ERS exercise can lead to a reduction of fat levels with no changes in muscle mass in elderly women. In other words, it can be presumed that ERS exercise improves mechanical muscle contraction ability, resulting in a change in the BMR, contributing to fat mass reduction, but not enhancing muscle mass due to age.”

................. On Line 359 to 361:

" Similarly, positive changes through ERS exercise from ANCOVA test in this study were observed in the trunk inclination and imbalance, pelvic tilt, kyphotic and lordotic angles, surface rotation to the right side, as well as lateral deviation to the left side after 12 weeks.”

We sincerely appreciate your comments,

because your comments make our manuscript better

Re-resubmission Date

29 October 2020

P/S

For reference, we send a document stating that the English language has been corrected.

Reviewer 2 Report

Many significant improvements have been made.

However, still it remains unclear, for which parameter sample size was calculated? Pain intensity?

In the limitation section it should be mentioned, that not credibility tests have been done in respect to the proper use of a horse simulator.

Author Response

Answers to 2nd reviewer’s comments

Thank you for your kind advice and comments for publication in Medicina. We revised our manuscript as per your comments. We represented the specific modifications in response to the comments by blue-letters in our manuscript. We sincerely appreciate your comments because your comments make our manuscript better.

#1. Comments or Suggestions

Many significant improvements have been made.

However, still it remains unclear, for which parameter sample size was calculated? Pain intensity?

In the limitation section it should be mentioned, that not credibility tests have been done in respect to the proper use of a horse simulator.

#1. Response:

In the present study, we calculated the sample size to estimate an adequate number of participants. As you know, samples should not be small, and should not be excessive in the scientific paper because too small a sample may prevent the findings from being extrapolated and too large a sample may amplify the detection of differences, emphasizing statistical difference (Altman, 1991). Therefore, an appropriate sample size is important for planning and interpreting scientific research. G*Power used in this study is a statistical power analysis software program, and is capable of performing a variety of power and sample size. We calculated an adequate sample size to reduce uncertainty and to increase the possibility to generalize the results induced in this study to the population of interest, although we mentioned a small sample size as the limitation of this study.

On your second comment ‘the proper use of a horse simulator’, we pre-trained participants how to ride the equine simulator, and researchers watched participants from the side and helped them to establish the correct posture during the ERS program. Therefore, participants could finish the ERS program safety and exactly during study period. According to your second comment, we inserted this explanation into the “2.2 Experimental design”.

For example,

2.2. Experimental design

The ERS program included 8 mins of warm-ups, 15 mins of exercise, and 7 mins of cool-downs. To prevent injuries, all participants underwent stretching while standing before the intervention, and learned how to ride the equine simulator. After boarding the simulator, participants walked for 3 mins as a pre-exercise before the actual exercise. During the ERS program, researchers watched participants from the side and helped them to establish the correct posture. After the actual exercise, participants walked again for 2 mins as a post-exercise and finished with stretching on the ground.

I wish this response can satisfy you, and appreciate your comment. Owing to your comment, I think that our study will be more improved.

[References]

Altman DG. Practical Statistics for Medical Research. London, UK; Chapman & Hall; 1991.

We sincerely appreciate your comments,

because your comments make our manuscript better

Re-resubmission Date

29 October 2020

P/S

For reference, we send a document stating that the English language has been corrected.

This manuscript is a resubmission of an earlier submission. The following is a list of the peer review reports and author responses from that submission.